# Multidrug-Resistant *Staphylococcus aureus* Strains Thrive in Dairy and Beef Production, Processing, and Supply Lines in Five Geographical Areas in Ethiopia

**DOI:** 10.3390/vetsci10120663

**Published:** 2023-11-22

**Authors:** Fikru Gizaw, Tolera Kekeba, Fikadu Teshome, Matewos Kebede, Tekeste Abreham, Halefom Hishe Berhe, Dinka Ayana, Bedaso Mammo Edao, Hika Waktole, Takele Beyene Tufa, Fufa Abunna, Ashenafi Feyisa Beyi, Reta Duguma Abdi

**Affiliations:** 1School of Veterinary Science, Arsi University, Assella P.O. Box 193, Ethiopia; fikru.gizaw@arsiun.edu.et; 2College of Veterinary Medicine and Agriculture, Addis Ababa University, Bishoftu P.O. Box 34, Ethiopia; tolerakekeba@gmail.com (T.K.); fikaduteshome@yahoo.com (F.T.); matewoskebede@gmail.com (M.K.); tabriham@yahoo.com (T.A.); halephom97@gmail.com (H.H.B.); dinka.ayana@aau.edu.et (D.A.); bedaso.mamo@aau.edu.et (B.M.E.); hika.waktole@aau.edu.et (H.W.); takele.beyene@aau.edu.et (T.B.T.); fufa.abunna@aau.edu.et (F.A.); 3Department of Veterinary Microbiology and Preventive Medicine, College of Veterinary Medicine, Iowa State University, Ames, IA 50011, USA; afbeyi@iastate.edu; 4Department of Biomedical Sciences, College of Veterinary Medicine, Long Island University, Greenvale, NY 11548, USA

**Keywords:** antimicrobial resistance, epidemiology, ecology, food safety, *S. aureus*

## Abstract

**Simple Summary:**

Monitoring and surveillance of food-borne pathogens and their antimicrobial resistance (AMR) in the food supply chain are fundamental for reducing food-borne hazards. *Staphylococcus aureus* is praised globally for readily acquiring AMR and causing foodborne illnesses. Studies are scarce on its reservoirs, hiding places, and routes of entry to the food supply chain in low-income countries. This study aimed to identify ecologies where *S. aureus* and its AMR strains are abundant in dairy and beef production and processing lines in five geographical areas in central Oromia in Ethiopia to guide whether food (meat and milk) is produced, handled, and moved in a safe and hygienic supply system. We found that *S. aureus* was prevalent, higher in dairy farms than in abattoirs. Its prevalence varied among 10 ecologies (10 sample sources) but did not vary among 5 locations. All isolates (100%) harbored AMR. The number of ineffective antimicrobial classes against them was high (range = 1–9; median = 5), indicating multidrug resistance (MDR) was prevalent. The abundance of MDR *S. aureus* varied between 5 locations and 10 ecologies, but the highest was in slaughter lines. Most isolates had different AMR patterns, indicating the isolates were phenotypically unrelated. We also detected some isolates with identical AMR patterns in different ecologies, suggesting their movement between ecologies or their ubiquitous presence in many ecologies. Overall, MDR *S. aureus* is abundant in a broader geographical area in central Oromia, contaminating milk, meat, equipment, and workers, demanding prompt regulations and operations on personnel safety, hygiene practices, and a comprehensive investigation of control.

**Abstract:**

Livestock, farms, abattoirs, and food supply systems can become the source of foodborne pathogens, including *S. aureus*, in the absence of monitoring, general hygienic practices, and control. Studies are scarce on reservoirs (hiding places) and routes of entry of *S. aureus* into the food supply chain in Ethiopia. To fill these gaps, we evaluated the role of cows (milk), meat, equipment, and food handlers on the abundance and AMR of *S. aureus* in five geographical areas in central Oromia, Ethiopia. We isolated *S. aureus* from 10 different ecologies per area in 5 areas and tested their sensitivity to 14 antimicrobials of 9 different classes. We ranked the 5 areas and 10 ecologies by computing their multiple AMR index (MARI) at a cut-off value of 0.2 to determine ‘high-risk’ ecologies for AMR. We recorded as MDR if an isolate had resistance to ≥3 antimicrobial classes. We used a circos diagram to analyze if isolates with identical AMR patterns were shared between different ecologies. *S. aureus* is prevalent in central Oromia (16.1–18.3%), higher in dairy farms than in abattoirs, and varied among 10 ecologies (*p* < 0.001) but not among 5 areas (*p* > 0.05). Of the 92 isolates, 94.6% were penicillin-resistant. Their AMR prevalence was above 40% for 9 of 14 antimicrobials. All isolates (100%) had AMR in at least one antimicrobial class (range = 1–9; median = 5), indicating MDR was prevalent. The prevalence of MDR *S. aureus* varied (*p* < 0.05) among areas and 10 ecologies; the highest was in slaughter lines. All isolates had a MARI of >0.2, indicating drug overuse, and *S. aureus’s* AMR burden is high in central Oromia. Dairy farms had higher MARI values (0.44) than abattoirs (0.39). Of 10 ecologies, the highest and lowest MARI values were in the beef supply chain, i.e., slaughter line (0.67) and butcher’s hand (0.25). Of the 68 different AMR patterns by 92 isolates against 14 antimicrobials, 53 patterns (77.9%) were unique to individual isolates, indicating they were phenotypically dissimilar. MDR *S. aureus* was widespread in central Oromia in dairy and meat supply chains, contaminating milk, meat, equipment, and workers in farm and abattoir settings. In the absence of strict regulations and interventions, MDR *S. aureus* can be disseminated from these epicenters to the public.

## 1. Introduction

*S. aureus* is an opportunistic commensal bacterium that causes contagious bovine mastitis [1], meat and milk contamination [2,3], and food poisoning by producing several toxins [4].

Owing to the absence of an effective vaccine, antimicrobials are widely used as a feasible intervention tool against *S. aureus* infection in low-income countries [5]. Antimicrobial overuse drives AMR emergence, posing public health insecurity globally [6,7,8]. In Ethiopia, antimicrobial misuse has been reported in medical [9,10], urban communities [11], and veterinary settings [12,13]. A functional multidisciplinary national AMR monitoring and tracking system, as well as any legal restrictions on veterinary antibiotic use, are lacking in Ethiopia [14]. High levels of AMR have been reported in multiple bacterial species in Ethiopia [15]. Hospital settings in Ethiopia are overwhelmed by MDR *S. aureus* isolates, and some strains are resistant to almost all antimicrobials in use [10,16,17,18]. The detection rate of MDR isolates from any species of bacteria is high in Ethiopian people, ranging from 43.5–75.9%, but it varies with geographical locations [9]. One of the pathogens with MDR is methicillin-resistant *S. aureus* (MRSA). The detection of MRSA isolates in humans is also high, ranging from 24.1–61% [18,19].

As food animals are the reservoirs of zoonotic and foodborne pathogens, imprudent veterinary antimicrobial use makes animals the epicenter of AMR by inducing selective pressure on pathogens [20,21,22]. Transmission of MRSA from livestock to humans has been reported between farm workers and animals [23,24]. This is of particular concern in Ethiopia since suitable factors exist for pathogen buildup in animals to subsequently disseminate to humans. The potential drivers of their transmission can be the substandard management of livestock in Ethiopia [25], the ubiquitous presence of animal feces over space [26], substandard abattoir management practices [19,27,28], and the habit of raw meat consumption [27]. The whole food supply chain, from farm to fork, is at risk of foodborne contamination in the absence of strong monitoring, surveillance, regulations, and general hygienic practices [28,29]. *S. aureus* is one of the foodborne contaminants that require intervention since it can enter the food chain through many routes [30]. To find an effective intervention, it must begin with a clear understanding of the population heterogeneities, reservoirs, and AMR of *S. aureus* along the farm-to-fork continuum, but information is scarce in Ethiopia. For that reason, this study investigated the role of milk, beef, equipment, and workers in dairy and beef production and processing lines on the abundance and AMR of *S. aureus* in five geographical areas in central Oromia, Ethiopia.

## 2. Materials and Methods

### 2.1. Study Area

The study was conducted in five cities in central Oromia, Ethiopia. They were Addis Ababa, Holeta, Bishoftu, Adama, and Assela. Addis Ababa is the capital of Ethiopia, with a human population of 3.4 million. Holeta is located 40 km west of Addis Ababa, with a human population of 25,000. Adama is located 99 km east of Addis Ababa, with a population of 300,000. Bishoftu is a tourist attraction city located 47 km southeast of Addis Ababa, with a population of 171,000. Assela is located 175 km east of Addis Ababa, with a population of 67,269. Sustainable water supply, solid waste management, and sanitation are major issues in the five cities [31].

### 2.2. Dairy Farms and Abattoirs: Sampling, Sample Types, and Sample Handling

The five cities were selected for three reasons: (i) the presence of intensive or semi-intensive dairy farms; (ii) the presence of municipal and/or private-owned abattoirs for beef-meat supply; and (iii) the high human population in the cities that are served by these dairy farms and abattoirs. Overall, 1001 samples from the five locations were collected: Assela (n = 181), Adama (n = 180), Bishoftu (n = 253), Addis Ababa (n = 193), and Holeta (n = 194). Using a simple random sampling technique, 514 samples were collected from 53 dairy farms in five locations. The samples included udder milk samples from cows, tank milk, buckets, farm tanks, milkers’ hands, and nasal swabs. The cows were zebu crossbred with Holstein and kept in an intensive or semi-intensive production system. Overall, 487 samples were collected from five abattoirs (one abattoir per location). The sample types were carcass, knife, slaughter line, butcher’s hand, and nasal swabs. Male zebu cattle were mainly slaughtered in the abattoirs.

A milk (2 mL) sample from each cow was collected after disinfecting the teats with 70% alcohol. Swab samples were also taken from the bucket, milk tanker, milker’s hands, and nasal area before the milking operation. In the abattoir, carcasses, knives, slaughter lines, personnel’s hands, and nasal swabs were taken from a 100 cm^2^ surface area using moistened swabs in buffered peptone water (BPW). The swab samples were put inside a sterile test tube containing 4 mL of BPW, capped, labeled, and stored in an ice-packed cool box. Finally, the samples were transported to the College of Veterinary Medicine, Addis Ababa University, and the Assela Regional Laboratory for bacteriological analysis.

### 2.3. Bacteriological Isolation and Identification

Non-selective pre-enrichment: An aliquot of 50 µL of each swab-BPW or milk sample was spread on tryptic soy agar with 5% sheep blood (Becton Dickinson Co., Sparks, MD, USA). The culture was incubated at 37 °C for 24 h. *S. aureus* colonies were evaluated using morphologies (round, smooth, and white or yellow colonies) and hemolytic patterns such as α, β, and double (α + β) hemolysis on the surface of the blood agar plate [32].

Plating out for identification by biochemical tests: Suspected *S. aureus* colonies were subcultured on nutrient agar plates to get a pure colony. Then it was incubated at 37 °C for 24 h. Final *S. aureus* identification was conducted using Gram-stain and biochemical tests including coagulase, catalase, indole production, methyl red test, Voges-Proskauer reaction, urease production, citrate utilization, and sugar fermentation [32,33].

### 2.4. Antimicrobial Susceptibility Testing

The susceptibility of the isolates was tested against 14 different antimicrobials (Oxoid, Hampshire, UK) of 9 different classes (Table 1).

The disk diffusion method was used by adjusting the turbidity of colonies to a 0.5 McFarland standard, cultured on Mueller Hinton agar, and placing antimicrobial-coated disks on the agar surface, followed by incubation aerobically at 37 °C for 18 h. Inhibition zone diameters were measured in millimeters to the nearest whole number, and the isolates were classified as resistant or susceptible to a specific antimicrobial based on the cut-off value as per the CLSI guidelines [34]. We identified MRSA isolates by testing against a cefoxitin disk (30 µg) because cefoxitin has been recommended as a reliable method for detecting MRSA strains [35,36]. We also screened isolates for MDR, and isolates that displayed resistance to ≥3 antimicrobial classes were considered MDR [37].

#### Multiple Antimicrobial Resistance Index (MARI) for Ranking Isolates and Ecologies

We ranked each isolate and its ecological origin (e.g., geographical locations, farms, abattoirs, and 10 different ecologies (10 sample types)) using their respective MARI value (i.e., based on their AMR burden). We calculated the MARI values of each isolate by:(1)MARI value=a/b,
where *a* is the number of antibiotics to which the individual isolate is resistant and *b* is the total number of antibiotics tested [38,39].

We also calculated the MARI value of different ecologies (location, abattoir, dairy farm, human, animal, and sample types) by applying the MARI indexing to each ecology (sample type/source), from which several isolates were taken using the formula:(2)MARI index of a given ecology (sample source)=a/(b×c)
where *a* is the aggregate antibiotic resistance score of all isolates from the sample source/type, *b* is the number of antibiotics, and *c* is the number of isolates from the sample source/type [38,40,41]. Since the MARI value is a good tool for health risk assessment, which identifies if isolates are from a region of high or low antibiotic use, we ranked the 5 areas and 10 ecologies by their MARI values. Ecologies with MARI values above 0.2 are considered ‘high-risk’ of antimicrobial overuse, indicating such ecologies have a high AMR burden [38,40,41].

### 2.5. Data Analysis

A descriptive analysis summarized the prevalence of bacterial isolates and AMR resistance. A comparison of the association between each potential risk factor (geographical location, sample source, and sample type) and *S. aureus* positivity was analyzed by the χ^2^ test (Fisher’s exact test). Univariate and multivariate logistic regression analyses were performed to measure the strength of the association between the potential risk factors and the positivity of *S. aureus*. We tested the hypothesis that the number of ineffective antimicrobial classes (response variable) against *S. aureus* was equal or not for all isolates originating from different geographical areas, sample sources, and sample types. The effect of these predictor variables (risk factors) on the response variable was tested/compared in two steps. First, the mean comparisons for their MARI values were compared by ANOVA. Second, multivariable linear regression was used to determine the predictors among the studied potential risk factors, such as area, sample type, and sample source. Finally, we used a circos diagram to analyze the relationship between ecologies and AMR patterns to test if isolates with identical AMR patterns were shared/circulated between different ecologies or if some unique AMR patterns exist in a particular ecology [42]. We used SPSS software (IBM SPSS Statistics for Windows, Version 20.0, Armonk, NY, USA) for data analysis and set the significance level at α = 5% and a 95% confidence interval.

## 3. Results

### 3.1. Prevalence of S. aureus

The prevalence of *S. aureus* in central Oromia was 17.2%, with no significant (*p* > 0.05) variation among the five locations, suggesting all the studied locations were equally affected. Its prevalence in dairy farms was significantly (*p* < 0.001) higher than in abattoirs. Among dairy farm sample types, *S. aureus* prevalence was significantly (*p* < 0.05) higher in tank milk than milkers’ nasal swabs and udder milk of cows. Among samples from the abattoirs, butchers’ hand and slaughter-line swabs had higher *S. aureus* contamination than carcass/meat swabs, indicating poor hygienic practices (Table 2). Finally, after controlling for other risk factors in multivariable logistic regression analysis, only the sample source had a significant (*p* < 0.000) effect on the positivity of *S. aureus*. Accordingly, abattoirs had lower *S. aureus* contamination than dairy farms (odds ratio for abattoir vs. dairy farms = 0.460; 95% CI = 0.326–0.650; *p* < 0.000), suggesting dairy farms are the epicenter for *S. aureus* abundance.

### 3.2. Prevalence of Antimicrobial Resistance

Penicillin resistance was the most widespread over large-scale ecological fronts (five study geographical areas and 10 sample types), followed by nalidixic acid, cloxacillin, and nitrofurantoin, among others, whereas the majority of the isolates were susceptible mainly to gentamicin, followed by ciprofloxacin, and kanamycin. Of the 92 *S. aureus* isolates tested, almost all of them (94.6%) had resistance to penicillin, indicating its widespread resistance, followed by nalidixic acid (76.1%), cloxacillin (66.3%), and nitrofurantoin (62.0%), among others. However, most of the tested isolates were susceptible to GEN (97.8%), CIP (89.1%), and KAN (88.0%) (Figure 1).

The prevalence of CIP-resistant *S. aureus* isolates from abattoirs (18.4%) was higher than CIP-resistant isolates from dairy farms (5.6%), in which most of the CIP-resistant isolates in abattoirs were from knife swabs (50%). The prevalence of CIP-resistant isolates was also higher in Addis Ababa, Adama, and Asella, but all isolates from Bishoftu and Holeta were susceptible to CIP. Although GEN was effective against most *S. aureus* isolates from almost all ecologies/sources, isolates from tank milk were unusually GEN-resistant (20%).

The prevalence of AMR isolates from milkers’ nasal swabs and tank milk was higher for many antimicrobials, i.e., 100% for AMP, CLX, NTF, and PEN, whereas those isolates from slaughter line swabs were 100% resistant to CFX, CLX, ERY, and PEN. The prevalence of AMR *S. aureus* isolates in different locations, personnel, food, and food-related equipment is displayed below, indicating AMR *S. aureus* is abundant in many ecologies (Figure 1).

### 3.3. MRSA vs. MSSA Isolate Detection by Cefoxitin (FOX) Disc

Of the 92 *S. aureus* isolates tested, the prevalence of MRSA isolates was 41.3%, and that of methicillin-susceptible *S. aureus* (MSSA) was 58.7%. Furthermore, 50% of the 92 isolates were resistant to 6.5 of the 14 different antimicrobials tested (median value = 6.5), but the maximum number of antimicrobials ineffective against many isolates was 4 (mode = 4; 20 isolates). Acquisition of the MRSA trait by *S. aureus* isolates (compared to the MSSA) was associated significantly (X^2^ = 36.61; *p* < 0.001) with an extreme AMR to many antimicrobials (i.e., MDR) ranging from 3 to 12 (mode = 7; median = 7.5), whereas the number of infective antimicrobials against MSSA isolates ranged from 1 to 8 (mode = 4; median = 4.5). Interestingly, 87.0% (47/54) of MSSA isolates had resistance to several antimicrobials without harboring the MRSA phenotype, suggesting other mechanisms may play a role in *S. aureus* for MDR in central Oromia (Figure 2A).

### 3.4. Multidrug Resistance (MDR)

Overall, 73.9% (68/92) of 92 isolates, 89.5% (34/38) of 38 MRSA isolates, and 63.0% (34/54) of 54 MSSA isolates showed MDR (resistance to ≥3 antimicrobial classes), indicating the MDR *S. aureus* population is widespread in many ecologies in Central Oromia (Figure 2B). Intriguingly, all isolates from the knife (n = 4) and 3 of 4 isolates from butchers’ hand (n = 4) were MSSA, whereas all isolates from the slaughter line (n = 4), 2 of 3 isolates from milker’s nasal swab (n = 3), and 4 of 5 isolates from tank milk (n = 5) were MRSA, all harboring resistance to more than five different antimicrobials (Figure 3). 

The number of antimicrobial classes to which a single isolate had resistance was counted, and their resistance ranged from 1 to 9 antimicrobial classes, as follows: 3.3, 22.8, 25.0, 19.6, 10.9, 9.8, 6.5, 1.1, and 1.1% of the 92 isolates had resistance to 1, 2, 3, 4, 5, 6, 7, 8, and 9 antimicrobial classes, respectively. Then we calculated the mean values of the number of antimicrobial classes ineffective in each of the four study areas, two sample sources, and 10 sample types and compared the means using ANOVA (Table 3; see the mean column). Only the mean difference of the 10 sample types was significant (*p* = 0.012), but the means of area and sample source were insignificant. For example, *S. aureus* from the Assela area had resistance against an average of 3.47 antimicrobial classes.

We also used area, sample source, and sample type as explanatory variables for predicting the number of antimicrobial classes ineffective in this study using a general linear model (r^2^ = 0.137; *p* = 0.022). After controlling the effect of all variables in the model, the number of antimicrobial classes ineffective against *S. aureus* was significantly higher in the Holeta area than at Assela (*p* = 0.029), as well as higher on butcher’s hand swab, bucket swab, milker’s hand swab, meat swab, and tank swab than in the udder milk of cows. The difference was not significant for isolates from milker nasal swabs, slaughter line swabs, and tank milk compared to udder milk, although the isolates from them had a higher AMR for many antimicrobial classes (i.e., 4–6 classes) than those from udder milk (Table 3).

### 3.5. MARI for “High-Risk” Source Determination

After noticing the presence of a large-scale widespread MDR *S. aureus* in the study area (Figure 4A–C), we computed MARI for each risk factor to determine ecologies with the “high-risk” source of contamination where many antimicrobials are used in the study area (Figure 4D–G). Accordingly, higher MARI values (high-risk contamination sites) than the 0.2 cut-off values were recorded for almost all isolates from various locations, abattoirs, dairy farms, and sample types, revealing many ecologies were contaminated with antimicrobial use (higher AMR). Isolates from slaughter lines (MARI = 0.67) had the highest MARI value, followed by milkers’ nasal swabs (0.52) and tank milk swabs (0.51), among others, but isolates from butcher’s hand swabs (MARI = 0.25) had the lowest, indicating the rank order of contamination status in the different ecologies. Overall, the MARI value in dairy farms (MARI = 0.44) was higher than in abattoirs (MARI = 0.39) (Figure 4E,G). All sample sources in this study had MARI values above the 0.2 cut-off value, which is an indicator of widespread contamination of AMR over a vast geographical area as well as high AMR contamination within a given location in diverse micro-ecologies (sample types) locally (Figure 4G).

### 3.6. Antimicrobial Resistance Pattern

The prevalence of isolates resistant to single, double, 3, 4, 5, 6, 7, 8, 9, 10, 11, and 12 antimicrobials of the 14 antimicrobials tested was 1.1, 6.5, 13.0, 21.7, 18.5, 14.1, 10.9, 5.4, 2.2, 3.3, 2.2, and 1.1%, respectively. The 92 isolates displayed 68 different AMR phenotypic patterns using 14 antimicrobials, suggesting *S. aureus* isolates in central Oromia are comprised of mixtures of heterogeneous strains. Of the 68 AMR patterns, 53 unique (dissimilar) AMR patterns were detected in 53 different isolates, suggesting the 53 isolates were phenotypically unrelated. The remaining 15 of the 68 AMR patterns were detected in the remaining 39 isolates, of which at least 2 may share one similar AMR pattern. Accordingly, we observed 8 dissimilar AMR patterns (each pattern carried by 2 isolates = 16 isolates), 6 dissimilar AMR patterns (each pattern carried by 3 isolates = 18 isolates), and 1 unique AMR pattern (CLX * NAL * PEN) (carried by 5 isolates). The 38 isolates from abattoirs exhibited 32 different AMR patterns, and the 54 isolates from dairy farms had 46 different phenotypic patterns. Five isolates showed the AMR CLX * NAL * PEN pattern, where two isolates were isolated from Butcher’s hand swab, two isolates from meat swab, and one isolate from tank swab, indicating similar isolates circulate in various ecologies or ecologies exchange isolates (Figure 5).

## 4. Discussion

*S. aureus* causes a significant economic and public health problem globally due to acquiring AMR [43,44,45]. However, information on *S. aureus* and its AMR in food supply chains is limited in Ethiopia due to the lack of a regular monitoring and surveillance system. This study investigated the prevalence of *S. aureus* and its AMR patterns in milk and meat, equipment, and humans in dairy farms and abattoirs in five locations in central Oromia, Ethiopia.

The prevalence of *S. aureus* was 17.5% in central Oromia, ranging from 16.1–18.3% across five locations in this study. The current finding was lower than the 40.6% in southern Ethiopia [46], 36% in northern Ethiopia [47], and 33.4% in Turkey [48]. The difference in prevalence can be due to spatiotemporal factors, antimicrobial use patterns among farms [49,50], and variations in hygienic practices on farms and in abattoirs [28]. The movement and admixture of different animals may also contribute to the disparity in the distribution of *S. aureus* strains [51].

The prevalence of *S. aureus* on dairy farms (22.4%) was significantly higher than in abattoirs (11.7%) in this study. This can be explained by the findings of other studies, which show that *S. aureus* is a principal pathogen of mastitis and a major reason for antimicrobial treatments in dairy cows [52]. A study on *S. aureus* in Turkey, however, reported that meat product samples (48.7%) had a significantly higher prevalence than milk and dairy products (23.2%) [48]. This suggests the ubiquitous presence of *S. aureus* in multiple food types.

*S. aureus* was isolated from udder milk (cows), tank milk, bucket swabs, and tank swabs. It was also isolated from personnel, equipment, and carcasses in the abattoir, indicating that *S. aureus* is a ubiquitous, highly resilient, and adaptive organism to different conditions. Other researchers have also reported *S. aureus* in bulk milk (75%), raw milk (37.8%) in Norwegian dairy farms [53], and raw milk (12.8%) in Iran [54]. The 22.6% prevalence of *S. aureus* in milk in this study was lower than the 40.6% in southern Ethiopia [46] and 36% in northern Ethiopia [47]. High *S. aureus* prevalence was reported among butcher shops, abattoirs, and street meat sales in northern Ethiopia (28.6–42.9%) [55], 37% in beef meat in the USA [56], and 29.41% in beef in Italy [57]. Identifying the sources of *S. aureus* and transmission pathways can help control its spread to the general public through food of animal origin. The contamination of food (milk and meat), personnel, and equipment by *S. aureus* in this study may be an indicator of substandard hygiene and food safety issues in the current study area. Care is needed during food production since body parts of animals such as hock skin, teat skin, groin, vagina, nares/muzzle, and other skin wounds are the source of *S. aureus* [58,59]. Not only animals but also personnel can be the source of *S. aureus*. In our study, we found that 20% and 23.5% of dairy farm workers (milkers) harbored *S. aureus* on their hands and in their nasal cavities, respectively, whereas 18.9% of abattor workers had it on their hands, but it was negative in the nasal swabs of abattoir workers. In this regard, about 20% of human populations are persistent *S. aureus* nasal carriers, 30% are intermittent carriers, and about 50% are non-carriers [60]. Up to 50% of personnel can potentially be the source of *S. aureus*.

### 4.1. Antimicrobial Resistance

In our study, none (0%) of *S. aureus* isolates were pan-susceptible. This is contrary to a pan-susceptibility of 6.9% in Inner Mongolia, China [61], 65.7% in Tennessee, USA [62], and 75% in Wisconsin, USA [63]. All isolates (100%) were resistant to at least one antimicrobial agent in this study, but the prevalence of *S. aureus* resistance to at least one antimicrobial agent was 93.1% in Inner Mongolia, China [61], and 34.3% in Tennessee [62]. For 9 out of 14 antimicrobials tested in this study, the AMR prevalence was high (above 40%), except for chloramphenicol (22.8%), sulphamethoxazole trimethoprim (15.2%), kanamycin (12.0%), ciprofloxacin (10.9%), and gentamicin (2.2%). It partly agrees with reports of AMR *S. aureus* in southern Ethiopia that ranges between 60.3–70.9% against oxacillin/methicillin, penicillin, and ampicillin; between 30.9–38.5% against amoxicillin-clavulanic acid, erythromycin, and vancomycin; 7.7% sulfamethoxazole-trimethoprim; and 0% ciprofloxacin [46]. The higher resistance to penicillin in our finding agrees with the world trend since *S. aureus* resistance to penicillin is increasing worldwide, such as 97.5% in humans in Iran [64], 92.7% in meat and dairy products in Turkey [48], 5–50% in Belgium, France, Latvia, Spain [65], and Italy [57,65]. However, the prevalence of AMR among isolates of *S. aureus* isolated from cattle is generally lower than 5% for several antimicrobials, including penicillin, in Scandinavia, the UK, the Netherlands [65], Canada [49,50], and the USA [62]. The variation by country may be due to variations in laboratory methods, reporting, prevention and control strategies, population densities [44], pharmaceutical regulatory reinforcement [14], differences in antimicrobial use/misuse at the farm level [50], and differences in food hygiene [28].

Gentamicin (2.2%) and ciprofloxacin (10.9%) resistances are low in this study. None (0%) of the isolates exhibited resistance to ciprofloxacin in southern Ethiopia [46]. Thus, both antimicrobials seem relatively effective against *S. aureus*. Little information exists on the use, prescription, or distribution of gentamicin and ciprofloxacin in Ethiopia for veterinary practice. Lack of use of both antimicrobials may be the reason for the lowered resistance of the isolates to them. Restricting veterinary use of some critically important antimicrobials in human medicine may delay or prevent the emergence of resistance, thus helping to prolong the efficacy of the antimicrobials for human use.

### 4.2. MDR and AMR Patterns in Different Sample Sources

In this study, ≥50% of all *S. aureus* isolates in central Oromia had resistance to five or more antimicrobial classes (i.e., the median number = 5) (Figure 1 and Figure 2). The prevalence of the MDR *S. aureus* population in Central Oromia is high: 73.9% (68/92) of 92 isolates, 89.5% of 38 MRSA isolates, and 63.0% of 54 MSSA isolates. Our finding is higher than the 62.8% reported in southern Ethiopia [46], 52% in meat samples in the USA [56], and 4.2% in milk in Tennessee [62]. Similarly, MDR isolates have been reported in beef in Italy [57] and in human and ovine isolates in France [66]. The probable reasons for higher MDR in our study area can be the misuse of antimicrobials, poor drug quality, inadequate surveillance, unhygienic conditions [14,67], and the intensity of antimicrobial consumption by cattle producers in the area [12,13,22]. The isolates in our findings displayed 68 AMR phenotypic patterns, which is higher than the 35 different AMR patterns observed in Iran [54] and 21 AMR patterns in the USA [62]. We did not investigate in this study whether each of the 68 different AMR phenotypic patterns is genetically different from each other or not. The ubiquitous presence of MDR isolates in multiple ecologies (sample types) in the food chain revealed critical substandard hygienic practices and food safety issues in the study area. A raw meat and milk consumption habit [27] coupled with food contamination by MDR *S. aureus* is a public health concern in Ethiopia.

### 4.3. Methicillin Resistance

Clinically, MRSA is the most important phenotype. The cefoxitin disk test has been regarded as a reliable predictor of the presence of the MRSA gene in *S. aureus*, i.e., *mec*A [35,36]. The cefoxitin test also coincides with MDR because a single genetic element confers resistance to the most commonly prescribed class of antimicrobials [44]. In this study, methicillin/cefoxitin resistance was 41.3%. The figure was lower than 60.3% of MRSA reports in southern Ethiopia [46], suggesting temporospatial variation in the prevalence of MRSA. Similarly, marked geographical variation in the prevalence of MRSA has been reported worldwide: 86.7% in Iran [64], >25% in 2008 in one-third of European countries [45], and >50% in 2009 in one-third of Latin American countries [68]. MRSA seems to be declining in Europe due to an increased understanding of its epidemiology [45]. In Ethiopia, however, the current findings and previous reports indicate that MRSA is widespread [18,46].

In our study, some isolates harbored MDR, although they were sensitive to cefoxitin (non-MRSA) (Table 3), suggesting the presence of other mechanisms of resistance in the area. The current finding suggests that gentamicin and ciprofloxacin seem effective in treating MDR isolates, including MRSA strains, but we did not verify this in in vivo trials.

## 5. Conclusions

The prevalence of *S. aureus* in central Oromia is 17.2%, but there is no significant variation between locations. Rather, within the five locations, some specific localities, such as dairy farms, are significant hotbeds for *S. aureus* abundance. *S. aureus* is also prevalent in raw meat, raw milk, equipment, and personnel who handle milk and meat. Accordingly, the reason for the abundance of *S. aureus* in all sampled ecologies seems to be poor general hygienic practices and the presence of reservoir animals and personnel. The majority of the *S. aureus* population (73.9%) is MDR (i.e., ≥3 antimicrobial classes). Phenotypically, these *S. aureus* populations had 68 different AMR patterns (subpopulations). The prevalence of MRSA is 41.3%. The presence of MRSA in food and food supply lines dictates the potential threat to public health in Ethiopia. Some personnel are carriers of *S. aureus* on their hands and nasals, indicating the hygiene of food handlers is important. Therefore, personnel safety, food hygiene, proper cooking, regular monitoring, and traceability of *S. aureus* phenotypes, genotypes, and AMR are recommended. Livestock and their products are reservoirs of MDR *S. aureus*, so controlling *S. aureus* infections in livestock can be one of the important steps to controlling *S. aureus* infections in humans.

## Figures and Tables

**Figure 1 vetsci-10-00663-f001:**
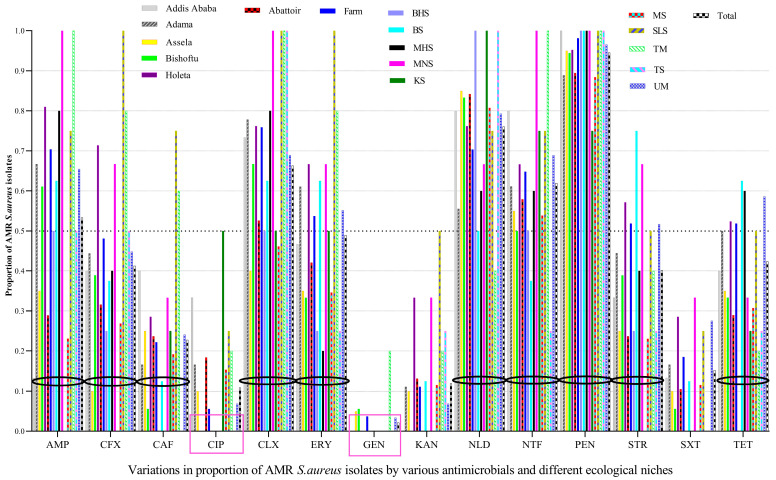
The prevalence of AMR in *S. aureus* varied by the types of antimicrobials tested, area, and sample type. AMR *S. aureus* isolates were abundant on butcher’s hand swabs (BHS), meat swabs (MS), bucket swabs (BS), milker’s hand swabs (MHS), tank swab (TS), milker’s nostril swabs (MNS), slaughter line swabs (SLS), udder milk (UM), knife swabs (KS), and tank milk (TM), which needs prompt interventions from a food safety perspective. GEN and CIP were effective antimicrobials against most *S. aureus* isolated from almost all ecologies/sources, with some exceptions. Black circles were used to demarcate the bars of a given antimicrobial from bars of another antimicrobial. Pink squares showed the two most effective antimicrobials against *S. aureus*.

**Figure 2 vetsci-10-00663-f002:**
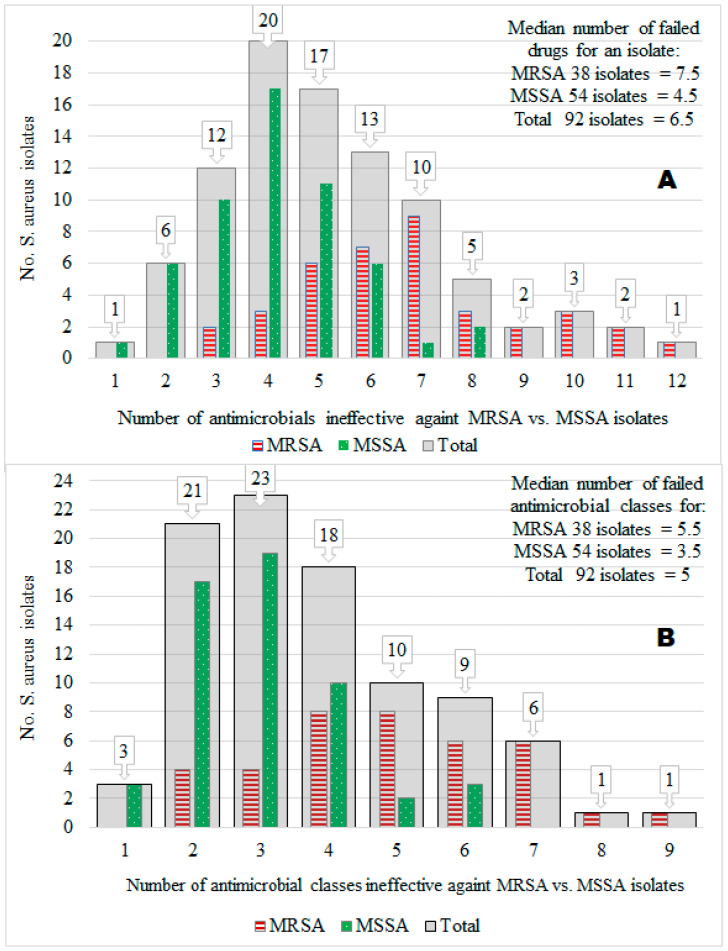
The number of isolates (*y*-axis) vs. the number of ineffective antimicrobials (*x*-axis) against the isolates by 14 antimicrobial types (**A**) and 9 antimicrobial classes (**B**). We showed this relationship for a total of 92 *S. aureus* isolates (gray color bars). We also showed it for 38 MRSA isolates (red color bars) and 54 MSSA isolates (green color bars).

**Figure 3 vetsci-10-00663-f003:**
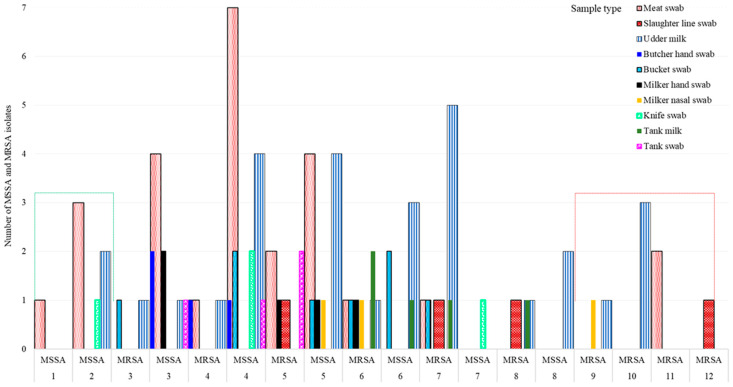
Variation in MSSA and MRSA *S. aureus* strain abundance by sample type and their susceptibility to antimicrobials. Note: Isolates that acquired MRSA became extremely resistant, ranging up to 12 antimicrobials.

**Figure 4 vetsci-10-00663-f004:**
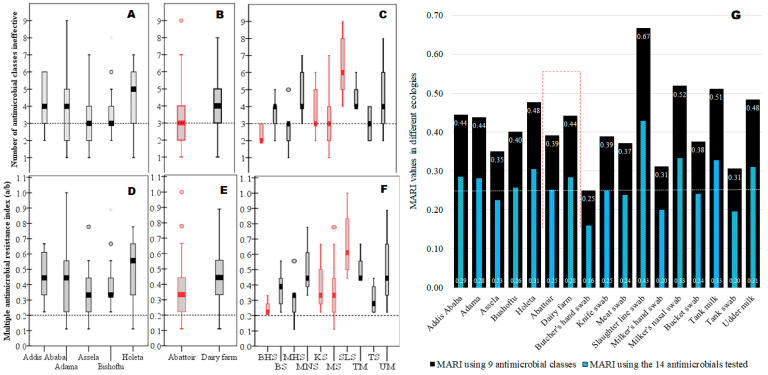
Ranking of various ecologies impacted by AMR *S. aureus* using their MDR and MARI values. Legend: The boxplot at the top shows the range of ineffective nine antimicrobial classes against each isolate as summarized by the median value (median line with error bars) for different geographical areas (**A**), abattoir and dairy farms (**B**), meat processing and milk production lines (**C**), including the cut-off value of 3 for MDR. The boxplot in the middle showed the MARI value computed by a/b for each isolate as summarized by median values (median line with error bars) for different geographical areas (**D**), abattoirs and dairy farms (**E**), and meat processing and milk production lines (**F**), including the cut-off value of 0.2 for MARI. The graph on the bottom (**G**) shows the MARI value computed by *a*/(*b* × *c*) to determine the “high-risk” source for the observed AMR. Keys: BHS = butcher’s hand swab; BS = bucket swab; MHS = milker’s hand swab; MNS = milker’s nasal swab; KS = abattoir knife swab; MS = meat swab; SLS = slaughter line swab; TM = tank milk; TS = tank swab; UM = udder milk (cow).

**Figure 5 vetsci-10-00663-f005:**
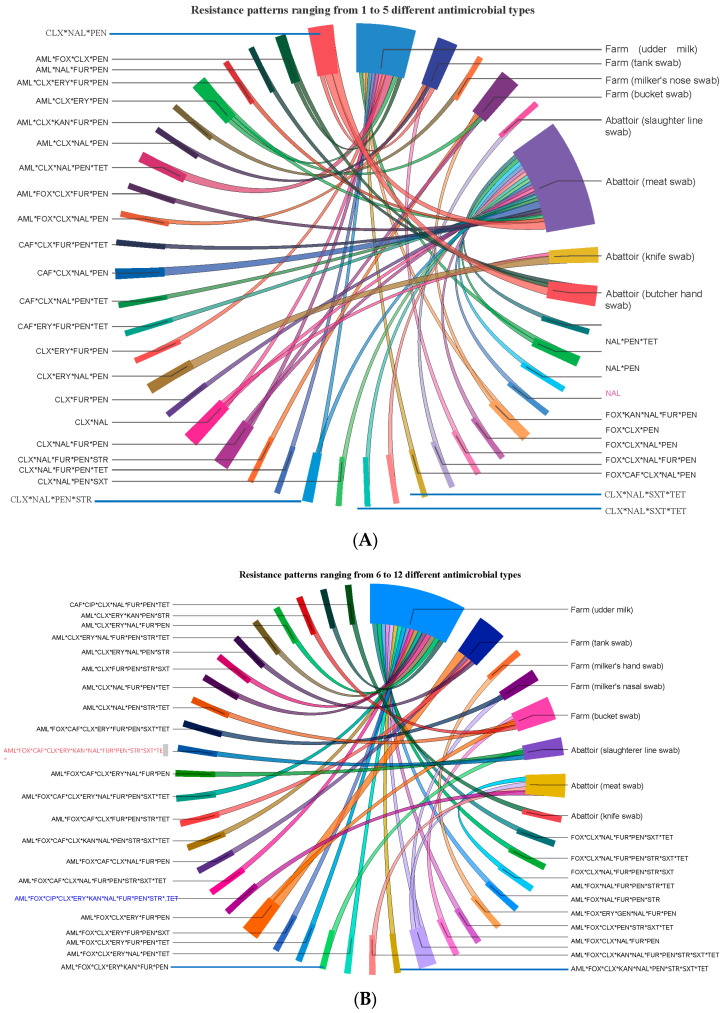
Ciscos diagram for showing the flow/source of 68 unique AMR patterns phenotypically produced by 92 isolates from food (meat and milk), personnel, and equipment on farms and abattoirs in Central Oromia. Isolates harbor resistance simultaneously to several antimicrobials, as shown here by their AMR patterns, namely, those with resistance to up to 5 antimicrobials (**A**) and those with resistance to 6 up to 12 antimicrobials (**B**).

**Table 1 vetsci-10-00663-t001:** Breakpoint zone diameter interpretive standards of 14 antimicrobials among 9 antimicrobial classes using disc diffusion susceptibility test.

Nine Antimicrobial Classes	14 Antimicrobial Agents (Potency)	Susceptibility Breakpoint (mm)
1. Aminoglycosides	Gentamicin, GEN (10 µg)	≥15
	Kanamycin, KAN (30 µg)	≥18
	Streptomycin, STR (10 µg)	≥15
2. Cephalosporins (b-lactams)	Cefoxitin, FOX (30 µg)	≥22
3. Folate pathway inhibitors	Sulfamethoxazole/Trimethoprim, SXT (25 µg)	≥16
4. Macrolides	Erythromycin, ERY (15 µg)	≥23
5. Nitrofurans	Nitrofurantoin, FUR (50 µg)	≥17
6. Penicillins	Amoxicillin, AML (25 µg)	≥20
	Cloxacillin, CLX (5 µg)	13
	Penicillin G, PEN (10 units)	≥29
7. Phenicols	Chloramphenicol, CAF (30 µg)	≥18
8. Quinolones	Ciprofloxacin, CIP (5 µg)	≥21
	Nalidixic acid, NAL (30 µg)	≥19
9. Tetracyclines	Tetracycline, TET (30 µg)	≥19

**Table 2 vetsci-10-00663-t002:** Prevalence of *S. aureus* in cattle-associated samples from abattoir and farm settings.

Origin	Sample Source	No.	Positive	Prevalence (%)	χ^2^	*p*-Value
Livestock setting	Dairy farm	514	115	22.4	20.00	0.000
	Abattoir	487	57	11.7		
Sample types from the abattoir	Butcher’s hand swab	37	7	18.9	26.32	0.03
	Slaughter line swab	37	7	18.9		
	Abattoir knife swab	37	5	13.5		
	Carcass/meat swab	361	38	10.5		
	Butcher nasal swab	15	0	0.0		
Sample types from the dairy farm	Tank milk	50	14	28.0		
	Milker nasal swab	17	4	23.5		
	Udder milk of cow	297	67	22.6		
	Bucket swab	50	10	20.0		
	Milker’s hand swab	50	10	20.0		
	Tank swab	50	10	20.0		
Total		1001	172	17.2		

**Table 3 vetsci-10-00663-t003:** The number (mean ± 95% CI) of antimicrobial classes ineffective against *S. aureus* using area, sample source, and sample type.

Variables	Mean No. of Antimicrobial Classes Ineffective		Generalized Linear Model for Predicting Infective Drug Classes Using Risk Factors				
Area	Mean	Std. Error	B	Std. Error	95% CI Lower	95% CI Upper	Sig.
Addis Ababa	3.94	0.374	−0.567	0.5007	−1.548	0.415	0.258
Adama	4.02	0.494	−0.483	0.5785	−1.617	0.651	0.404
Assela	3.47	0.367	−1.04	0.4761	−1.972	−0.106	0.029
Bishoftu	3.75	0.418	−0.750	0.4676	−1.666	0.167	0.109
Holeta	4.50	0.396	Ref ^a^	−	−	−	−
Sample source							
Abattoir	4.71	0.727	1.542	1.047	−0.511	3.595	0.141
Farm	3.17	0.383	Ref ^a^	−	−	−	−
Sample type							
Butcher hand swab	2.25	0.702	−3.71	1.09	−5.844	−1.57	0.001
Bucket swab	3.38	0.625	−1.00	0.449	−1.883	−0.125	0.025
Milker hand swab	2.80	0.724	−1.51	0.627	−2.740	−0.282	0.016
Milker nasal swab	4.67	1.157	0.658	1.101	−1.500	2.817	0.550
Abattoir knife swab	3.50	0.858	−2.10	1.195	−4.443	0.240	0.079
Meat swab	3.35	0.613	−2.47	1.052	−4.532	−0.409	0.019
Slaughter line swab	6.00	0.803	1.66	0.857	−0.049	3.359	0.057
Tank milk	4.60	0.598	0.252	0.4583	−0.646	1.151	0.582
Tank swab	2.75	0.758	−1.41	0.6414	−2.662	−0.148	0.028
Udder milk(cow)	4.34	0.520	Ref ^a^	−	−	−	−
Total	3.94	0.251					

Ref ^a^ = reference.

## Data Availability

Data will be made available on request.

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
