# Peer review of "Multidrug-Resistant Staphylococcus aureus Strains Thrive in Dairy and Beef Production, Processing, and Supply Lines in Five Geographical Areas in Ethiopia"

_vetsci, 2023, doi:10.3390/vetsci10120663_

Round 1
Reviewer 1 Report
Comments and Suggestions for Authors
Could you please provide an insight into possible strategies to monitor and mitigate/control MRSA (and in general AMR) spread in Ethiopia, please?
For example: to enhance the capacities of the national veterinary services, to strenghten the partnership between the local authorities and the universities, to provide guidelines to the stakeholders (farmers associations, etc), to provide an equitable access to vaccinations, etc
Author Response
Dear reviewer,
We have found that your comments are valid and attached our responses.
Thank you so much for devoting your time to improving our paper.

Reviewer 2 Report
Comments and Suggestions for Authors
The authors aimed to study the abundance of S. aureus multidrug resistant strains in dairy and beef production processing and supply lines in five geographical aeras. The authors should precise in the title that the five geographical aeras are in Ethiopia. The introduction provides a clear background to the study supported by references. The objectives of the paper are clearly defined, the methods are appropriate and data analysis is relevant. The results are particularly well commented and well illustrated, which makes them easier to follow. The conclusions are supported by the results and the discussion allows for the comparison of results for S. aureus in Ethiopia with those of other countries and practices.
Author Response

(The authors gave the same response as above.)

Reviewer 3 Report
Comments and Suggestions for Authors
This study by Gizav et al was conducted in Ethiopia. Samples were collected from five cities, including Addis Ababa, the capital city. Each of them has dairy farms and abattoirs. The samples were selected carefully: included udder milk from cows, tank milk, buckets, farm tanks, milkers' hands, and nasal swabs, n=514 in total (from 53 farms). Similarly, the abbatoir samples (n=487 from 5 places) included carcass, knife, slaughter line, butcher's hand, and nasal swabs.
Questions and comments
The identification of S. aureus could have been done with more sophisticated methods, at least using s. aureus Chromagar or rather a PCR-based method, if not Maldi-Tof.
Antibiotic susceptibility testing: line 147 “per the manufacturer’s recommendation.” This was not the case, according the cited reference, the authors have used the CLSI guidelines. Change the wording accordingly.
Line 165: “c is the number of isolates from the sample” – what does this mean? How many colonies were subcultured from a specific sample or what? Please explain. One of the cited reference (ref 38) is far too old to use for a methodology. Beside, I find this MARI calculation quite odd and unnecessary.
Table 1: give the full name of the antibiotics. Vancomycin should be included in testing.
Table 2: It reveals very interesting data about the prevalence of S. aureus from the different sources. Unfortunately the number of sampled human beings was a bit low, but still the nasal carriage rate was found to be 23.5% among milkers, which correlates well with international data. On the other hand, what could be the explanation that none of the tested 15 butchers carried S. aureus in their nostrils?
Page 6: do not use abbreviations like GEN or CIP in the flow text.
Figure 1 is far too complex and it is difficult to understand. It should be separated into several different figures.
Line 242: Why only 92 S. aureus out of the collected 172 isolates was tested for MRSA? Furthermore, a PCR should be made whether the MRSA isolates carry the mecA gene, or maybe mecC?
Figure 4: although it looks very fancy, but this is unnecessary as it provides too much information.
Lines 374-375: should be removed.
Far too many references were cited for a simple study like this.
In general, this manuscript provides quite interesting data about the prevalence of S. aureus and resistance of the isolates from different sample types from dairy farms and abbatoirs. However, the study did not apply any molecular method, only disc diffusion susceptibility testing; and the calculations about antibiotic resistance are overcomplicated. The figures are rather oddly prepared as well.
Comments on the Quality of English LanguageThe English of the manucripts is mostly very good.
Author Response
Dear reviewer,
We appreciate your critical thinking. We have found that your comments are valuable and attached our responses accordingly.
Thank you so much for devoting your time to improving our paper.

Reviewer 4 Report
Comments and Suggestions for Authors
The authors have isolated S. aureus from dairy and beef production in Ethiopia. The isolates were tested against different antimicrobials by disk diffusion. Resistance was alarmingly high. However, concerning the epidemiology of the isolates, the most important questions remain open.
Major remarks
MDR and MARI values were computed from the resistance data. MDR is evaluated against antibiotic classes and this was correctly done here. However, according to the definition of Magiorakis (doi: 10.1111/j.1469-0691.2011.03570.x ) for S. aureus, all betalactams with the exception 5. generations cephalosporins fall into one class, because one gene, mecA, will convey resistance to all betalactams with exception of the above mentioned 5. gen. cephalosporins. Otherwise, every MRSA will be resistant to two antibiotic classes with one gene, as seen in Fig. 3 and this skews the calculation of MDR. Please correct percentage of MRSA.
In addition, please give some information about the antibiotics that are used by veterinarians and human doctors in Ethopia; e.g. is nalidixic acid used? Are there any legal restrictions on veternary antibiotic use in Ethopia? Why wasn’t vancomycin or linezolid tested? These antibiotics would be active against nearly all isolates.
The sampling of the supply chains was designed to detect transmission of S. aureus and this is very interesting from an epidemiological point of view. To this end, the authors compared the resistance patterns. Resistance patterns varied strongly, which speaks against horizontal transmission of these isolates. However, using resistance phenotypes is not state of the art staphylococcal epidemiology. Normally the isolates are spa typed. The reason is that actually most of the strains are MRSA in this study and MRSA belong to only a few defined lineages, that acquire and lose resistance genes and often are host specific. Therefore, transmission pathways would be much clearer after spa typing and this would make the paper interesting for a wider readership.
Minor remarks
Singular: bacterium and plural bacteria
line 404: gentamicin
livestock is
Table 1: Please explain abbreviations.
line 217: How were samples tested for antibacterial resistance chosen? How did you avoid – if testing multiple samples from one source – to test the same strains twice?
Fig. 1: What is the significance of the circle?
line 241: Most importantly, cloxacillin resistant strains are MRSA, therefore, there were at least 66% of MRSA.
line 261: please give numbers of strains
line 272: resistance against
Fig. 4: Please explain colours.
line 373: sawbs must be swabs
line 388: resistance to the agents here can be conveyed by mecA
Author Response
Dear reviewer,
We appreciate your critical thinking. We have found that your comments are valuable. Our responses to clarify things and address your concerns are attached.
Thank you so much reviewer for devoting your time to improving our paper.

Reviewer 5 Report
Comments and Suggestions for Authors
In the manuscript, Gizaw et al. attempted to monitor the epidemiological and multidrug-resistant features of S.aureus in the food supply chain. The preliminary data could be considered as a fundmental step for S.aureus control in the area. My suggestions are as follows,
(1) Line 172-177 Such sentences do not belong to data analysis section. Please erase them.
(2) Line 101 There should be a blank between 40 and km. Line 120, 2ml should be 2 mL.
(3) "P" value shoud be uppercase and italic. Please modify all of them in the context.
Comments on the Quality of English Language
Minor editing of English language required.
Author Response
Dear reviewer,
We have addressed your comments and attached our responses.
Thank you so much reviewer for devoting your time to improving our paper.

Round 2
Reviewer 4 Report
Comments and Suggestions for Authors
This is the first revised version of the manuscript.
Please note that the authors did not specifically answer the comment to typing, however, argue that it is not possible to import PCR primers into Ethiopia. And yes, MRSA detection would also be better using PCR:
MRSA is defined as a strain that acquired one of the mec genes and shows resistance to penicillinase-stable betalactams. According to CLSI and CDC you can either check resistance to oxacillin on Mueller-Hinton agar containing 4 % NaCl at 33-35 °C for 24 h. Cloxacillin is distinguished only by one additional Cl atom from oxacillin and belongs to the same class like methicillin and oxacillin. This means that growth in the presence of cloxacillin indicates presence of an MRSA, also if the test was not performed correctly according to CLSI (salt missing in MH, temperature too high and incubation too short) or by cefoxitin screening. The MRSA percentage of the authors is derived from the cefoxitin screening but there are about 20 % more positive hits using the cloxacillin disks. As the authors work with animal isolates and cloxacillin is a veterinarian drug, this might be plausible. All strains that show either resistance to cefoxitin or cloxacillin are phenotypic MRSA. Any doubts should be resolved by PCR. Therefore the number of suspected or phenotypic MRSA should comprise all cloxacillin and/or cefoxitin resistant strains.
Author Response
Dear reviewer,
We appreciate you once again for dedicating your expertise to the improvement of our paper as well as to advance science. Thank you for your energy.
We specifically addressed the concerns of the reviewer on typing.
Concerning cloxacillin, we refrain from using cloxacillin resistance as a predictor of MRSA since cloxacillin is not standardized for MRSA detection in Gram-positive bacteria (attached file). Cloxacillin, instead, is standardized for detection of AmpC beta-lactamases, ESBL, and carbapenemases in Gram-negative bacteria. To our knowledge, cefoxitin and oxacillin +4% NaCl are the most reliable, sensitive, specific, and standardized for MRSA detection so far.
Thank you!
